# Improving Psychological Comfort of Paramedics for Field Termination of Resuscitation through Structured Training

**DOI:** 10.3390/ijerph18031050

**Published:** 2021-01-25

**Authors:** Chungli Bang, Desmond Ren Hao Mao, Rebacca Chew Ying Cheng, Jen Heng Pek, Mihir Gandhi, Shalini Arulanandam, Marcus Eng Hock Ong, Stella Quah

**Affiliations:** 1Acute & Emergency Care Department, Khoo Teck Puat Hospital, 90 Yishun Central, Singapore 768828, Singapore; mao.desmond.r@ktph.com.sg (D.R.H.M.); bluevoliet@gmail.com (R.C.Y.C.); 2Emergency Department, Sengkang General Hospital, Singapore 544886, Singapore; pek.jen.heng@singhealth.com.sg; 3Biostatistics, Singapore Clinical Research Institute, Singapore 138669, Singapore; mihir.ghandi@scri.cris.sg; 4Centre for Quantitative Medicine, Duke-NUS Medical School, Singapore 169857, Singapore; 5Emergency Medical Services Department, Singapore Civil Defence Force, Singapore 408827, Singapore; shalini_arulanandam@scdf.gov.sg; 6Health Services & Systems Research, Duke-NUS Medical School, Singapore 169857, Singapore; marcus.ong.e.h@singhealth.com.sg (M.E.H.O.); stella.quah@duke-nus.edu.sg (S.Q.); 7Department of Emergency Medicine, Singapore General Hospital, Singapore 169608, Singapore

**Keywords:** termination of resuscitation, psychological comfort, paramedics, Asian, training for termination of resuscitation

## Abstract

This study examines the impact of a newly developed structured training on Singapore paramedics’ psychological comfort before the implementation of a prehospital termination of resuscitation (TOR) protocol. Following a before and after study design, the paramedics underwent a self-administered questionnaire to assess their psychological comfort level applying the TOR protocol, 22 months before and one month after a 3-h structured training session. The questionnaire addressed five domains: sociocultural attitudes on resuscitation and TOR, multi-tasking, feelings towards resuscitation and TOR, interactions with colleagues and bystanders and informing survivors. Overall psychological comfort total (PCT) scores and domain-specific scores were compared using the paired t-test with higher scores representing greater comfort. Ninety-six of the 345 eligible paramedics responded. There was no statistically significant change in the mean PCT scores at baseline and post-training; however, the “feelings towards resuscitation and TOR” domain improved by 4.77% (95% CI 1.42 to 8.13 and *p* = 0.006) and the multi-tasking domain worsened by 4.11% (95% CI −7.82 to −0.41 and *p* = 0.030). While the structured training did not impact on the overall psychological comfort levels, it led to improvements in the feelings of paramedics towards resuscitation and TOR. Challenges remain in improving paramedics’ psychological comfort levels towards TOR.

## 1. Introduction

In suitable patients, field termination of resuscitation (TOR) by paramedics has been shown to be effective in reducing conveyance of futile resuscitations to the hospital [1], conserving limited resources [2] and averting dangerous high-speed transports [3]. TOR protocols have been practiced in Canada and the United States of America [2] for more than 20 years, and there is a trend towards worldwide adoption of such TOR protocols. Despite this, prehospital TOR protocols have not been adopted in many Asian countries for various reasons. These include differences in the sociocultural context surrounding death [4] and legal barriers [5]. The “rescue culture” mentality and the perception of death as an unsuccessful outcome of many prehospital emergency medical services (EMS) [6] also limit the uptake for any TOR protocol.

There are inherent challenges to the application of the TOR protocol itself. Applying TOR protocols involves the fulfilment of one pre-requisite and two components [5,7] (Figure 1). The pre-requisite is the performance of cardiac resuscitation on the patient, while the two components include assessing for suitability of TOR application based on pre-determined criteria and the breaking of bad news to the patient’s loved ones. The complexity of the pre-requisite and the two components may be a source of stress to the paramedic. The negative impacts on practitioners such as stress and burnout that arise from death pronouncement and breaking bad news have been well documented [8]. Prior research showed that terminating resuscitation poses difficulties to paramedics from external factors such as perceived societal roles of paramedics [9] and challenges in executing the protocol [10] and internal factors such as those governed by personal emotions and beliefs. Furthermore, the lack of formal training in breaking bad news has been a barrier to the uptake of the TOR protocol [11].

The five key domains that make up the construct of paramedics’ psychological comfort with field pronouncement of death have been previously described and validated in Morrison’s questionnaire [12]. These domains include sociocultural attitudes on resuscitation and TOR, multi-tasking, interactions with colleagues and bystanders, feelings towards resuscitation and termination of resuscitation, and informing survivors [12].

The pre-requisite of training for cardiac arrest scenarios is common and fundamental to most EMS training systems in the world [13]. However, there has been a lack of focus in Singapore EMS training on the latter two components addressing the communication and the education in applying the TOR protocol (Figure 1). Training to improve communication skills and confidence in breaking bad news has been shown to be essential [14] and previously described in a single pilot study to have the potential to improve paramedics’ psychological comfort with TOR [11]. The second component of TOR training addresses the understanding of the basis of the TOR protocol: the science behind TOR criteria and the assessment of patient suitability. This gap in TOR training has not been previously described in the literature.

A 3-h structured training session was designed to focus on the latter two components of TOR training as the pre-requisite had been covered in the core syllabus of Singapore paramedic education. This study is the first in Singapore to quantitatively assess the impact of structured training involving coaching in breaking bad news and the education of the basis of TOR in improving paramedic’s psychological comfort. 

## 2. Material and Methods

### 2.1. Study Setting and Participants

The study was conducted in Singapore, a Southeast Asian island city-state nation of 5.7 million (2019) people [15], with approximately 345 certified paramedics from the Singapore Civil Defence Force (SCDF) and private operators responding to EMS calls from a single national emergency number [16]. There were 191,468 EMS calls made in 2019 [17] and 5094 deaths that occurred in residence in 2018 [18]. Prehospital care is well established with a tiered emergency medical service (EMS) framework, and paramedics are from various ethnic, social, cultural and religious backgrounds. The paramedics are fluent in English and undergo an 18-month training in paramedicine before licensure and practice. 

### 2.2. Patient and Public Involvement

No patients or members of the public were involved in the design, conduct, reporting or dissemination plans of this research.

### 2.3. Study Design

This is a before and after study of paramedics’ psychological comfort (Figure 2 and Figure 3). The pre-training test was conducted 22 months before a 3-h structured training session on TOR (Figure 2). The post-training test was carried out one month after the structured training session.

### 2.4. Training

The 3-h training session had two pedagogical objectives:(a)To impart skills in breaking bad news;(b)To provide scientific evidence for the TOR protocol.

Results from a baseline questionnaire [16] were used to guide the structured training. Emphasized domains included “feelings towards resuscitation and termination of resuscitation”, “sociocultural attitudes on resuscitation and TOR” and “informing survivors”. Various formats and delivery methods including online videos, lectures, role-play simulations and group learning sessions were utilized (Table 1). (* Summary of training content provided in Appendix B).

### 2.5. Instrument

The original Morrison questionnaire consists of 22 questions measuring the construct of “psychological comfort with field pronouncement of death” [12]. Responses to the close-ended questions in the Morrison questionnaire are designed as a 5-point Likert scale response questionnaire (5 = never to 1 = always). Morrison developed her 22-item questionnaire based on data from focus group discussions with paramedics and feedback reviews from 41 stakeholders. Six questions were added to the original Morrison questionnaire to address four possible additional predictors of psychological comfort: knowledge of survival probability, religious affiliation, the location of the patient and perceived trust of the family. These four factors are pertinent to the local sociocultural context in Singapore.

The questionnaire (Box 0) had shown good construct validity and had been revalidated in the local setting with a good Cronbach alpha score (α = 0.896) and moderate test–retest reliability (r = 0.627) [16]. The psychological comfort total (PCT) score was the sum of the total responses to 28 questions with higher scores representing greater comfort. Both PCT score and individual domain scores were re-scaled into a percentage score (0 to 100), with a higher score representing greater comfort for easy interpretation.

Sample size: We postulated that overall PCT scores would improve by a moderate effect size following the training. A sample size of 90 participants was required to detect a difference of 0.3 SD (moderate effect size) in the overall PCT score to achieve 80% power with a 5% (two-sided) level of significance [19]. The effect sizes of the five domains were calculated.

### 2.6. Data Collection

Both the pre-test and post-test were conducted as self-administered questionnaires distributed to the paramedics during their compulsory monthly educational sessions. Informed consent was obtained as participation in the questionnaire was voluntary. Missing data were presumed to be at random (<5%) and were imputed using the median value of other questions in the same domain.

### 2.7. Data Analysis

The pre-test and post-test data were compared using the paired *t*-test and the 95% confidence interval (CI). A *p* value of less than 0.05 was considered statistically significant. No multiplicity correction was conducted considering the exploratory nature of the study.

The analysis comprised the following assumed predictors of psychological comfort: personal characteristics (age, gender, religious affiliation, previous personal experience with death and dying); and professional experience (years of professional experience as a paramedic, number of out-of-hospital cardiac arrests responded to in the past year and employment type (public vs. private sector)). Further regression analysis was performed on the change in overall PCT percentage to the respondents’ characteristics.

## 3. Results

### 3.1. Demographics

Ninety-six out of 345 eligible paramedics (27.8%) responded. Approximately half of the respondents were male, and a majority (85.4%) professed a religion. The mean experience of the respondents as practicing paramedics was eight years. Most participants (96.9%) had at least one death pronouncement in an out-of-hospital setting cardiac arrest in the previous 12 months (Table 2). Regression analysis of the change in overall PCT percentage to respondents’ characteristics demonstrated no statistically significant results apart from “age”, which had a *p* value of 0.049 (analysis provided in Appendix A).

### 3.2. Main Study Outcomes

Moderate effect size (0.356) was observed for the “feelings towards resuscitation and TOR” domain and the remaining domains yielded small effect sizes with their absolute numbers smaller than 0.3 (Figure 4). The overall PCT score at baseline and post-training did not show any statistically significant change: mean change = −0.58%, 95% CI (−2.73 to 1.53), *p* = 0.576 (Table 3). The PCT score in the feelings towards resuscitation and TOR domain increased by 4.77%, 95% CI (1.42 to 8.13), *p* = 0.006, while the PCT score in the multi-tasking domain decreased by 4.11%, 95% (−7.82 to −0.41), *p* = 0.030 (Table 3).

## 4. Discussion

This study showed an insignificant change in paramedics’ psychological comfort scores following a structured 3-h training as measured by the Morrison questionnaire. The statistically significant change in the psychological comfort based on the respondents’ age did not yield any meaningful impact as this was a paired longitudinal study where the post-training responses were compared with the initial responses. While the training improved feelings towards resuscitation and the need for a TOR protocol, there was a decrease in the multi-tasking domain PCT scores. When dissected, three out of five domains did not show a significant change. The varying levels of emphasis on different targeted domains may have contributed to this finding (Table 3). There was a lower level of emphasis on the “interaction with colleagues and bystanders” domain; therefore, it was unsurprising that this domain’s post-training score change was insignificant.

The training had attempted to address the other two domains of “sociocultural attitudes on resuscitation and TOR” and “informing survivors”. Both domains relate primarily to the confidence in communicating with the patient’s family on field TOR and death notification. Studies have shown that breaking bad news training increased healthcare workers’ comfort in accepting death [11,20]. However, this analysis did not replicate such an outcome. Compared to other healthcare settings, Singapore paramedics face a different set of challenges such as a time-constrained communication due to operational demands, legal constraints in death pronouncement such as non-binding “do-not-resuscitate” orders and differing public expectations regarding transport to the hospital for further resuscitation [21]. In light of these circumstances, a single structured training session may have been insufficient to increase the paramedics’ confidence when communicating with families. The repeated questionnaire was performed before the actual implementation of TOR, limiting the ability to see the impact of structured training on actual practice.

As expected, education on the evidence for TOR led to an improved score in the “feelings towards resuscitation and TOR” domain. Part of the difficulty in adopting such protocols has been the “rescue mentality” of EMS. It is demonstrated in this study that perceptions towards death and TOR could change when the scientific evidence behind the protocol is thoroughly understood. The training design via didactic lectures was able to achieve the transmission and acceptance of this information.

The outcome in the multi-tasking domain was unexpected as the domain score decreased after the structured training. However, the *p* value in the non-parametric Wilcoxon signed-rank test was insignificant compared to the paired *t*-test. A statistically significant finding in this domain could be a false positive as a significant decrease in the score was not expected. Furthermore, the non-clinically meaningful effect size for this domain could explain the above finding.

Questions in this domain centered around core paramedic skills such as application of automated external defibrillator pads. The study participants were reasonably experienced, and a majority (61.5%) had attended to six or more cardiac arrests in the past 12 months. It is unlikely that this finding represents a reduction in the actual core resuscitation skills. A probable explanation for this finding could be the cognitive overload experienced by paramedics in executing a complicated multi-step cardiac arrest protocol that now incorporates an additional component of TOR. Adding to the cardiac arrest protocol’s complexity, the need to remember and execute additional steps could have led to increased stress as participants feared that routine procedures were now comparatively neglected. Where appropriate to the setting, efforts should be made by policymakers to simplify TOR protocols. This finding deserves to be explored and replicated in further studies.

Educational programs that focus on breaking bad news differ in length and format but share similar components such as didactic teaching, role-playing or simulation [22], group discussion [23] and viewing of instructional videos [24]. The literature suggests that such training programs have been effective in improving the delivery of news and practitioners’ confidence [25]. Beyond the format, this study highlights the unique needs of training for TOR. While the ability to break bad news is a key component, other factors such as multi-tasking and addressing feelings towards the TOR protocol need to be similarly addressed. There is no “one size fits all” approach to TOR training, and it is imperative to understand and clarify the learning needs so that training programs can address these needs accordingly.

Prehospital educators ought to be clear-eyed regarding the utility of short single-session training. Some studies have suggested that continuing medical education may be more important than a single session for practitioners to gain the ability to internalize and execute the protocol smoothly [26]. Further training sessions should take into account feedback from paramedics and could take the form of specific case discussions and role-plays.

Lastly, it is possible that scores could be affected by an active “run-in” period following implementation. With practice, the informing survivors and multi-tasking domains could see some improvement as paramedics gain experience and confidence in the execution of the protocol. Future studies should address the effect of a run-in period and evaluate the need for continual medical education.

In summary, this study provides new information about the usefulness and limitations of structured TOR training and can inform settings that are considering introducing similar protocols. In the implementation of a new TOR protocol, factors other than paramedics’ psychological comfort should be considered, and these include the complexity of the protocol itself [7], public expectations of not dying at home [4] and legal barriers towards field death pronouncement by paramedics [5,27].

## 5. Strength and Weakness

The strength of the study lies in its longitudinal nature that allows a sequential observation of the changes after an educational intervention in Singapore paramedic participants. As mentioned in the discussion, this study was conducted before the actual implementation of the TOR protocol and the findings may change significantly once the paramedics have real-life experience with the application of the TOR protocol.

The response rate (27.8%) was broadly in line with other studies of paramedics’ attitudes and perceptions on clinical trials [28,29]. Historically with all researchers who employ questionnaires, many studies are confronted regularly with the issue of non-responders and its impact on conclusion inferences. The low response rate in this study was a significant limitation and may predispose the questionnaire’s outcome to non-response bias. The sample size determination strategy adopted in this study partially mitigated the effect of non-response bias.

Furthermore, the length of this study spanned almost two years with a one-year interval between the pre-test questionnaire and the training, and confounding factors such as the time sequence effect, the turnover of paramedic staff and the voluntary nature of the self-administered questionnaire may have influenced the response, thus leading to an unanticipated outcome.

## 6. Conclusions

The structured training program is the first of its kind in Singapore and, to our knowledge, in Asia that addresses the gaps in the training of TOR application. This study has uncovered many contextual challenges and the limitations of such approach in improving paramedics’ psychological comfort levels towards TOR. While structured training did not significantly impact the overall psychological comfort levels, it led to improvements in feelings towards resuscitation and TOR of paramedics. TOR protocols may lead to cognitive overload, thereby leading to a reduction in confidence in multi-tasking.

The inherent design challenges of a questionnaire, compounded with the training program’s heterogeneous nature and the study’s voluntary nature, limit a robust conclusion to be drawn. Nevertheless, this approach has shown a positive impact on paramedics’ psychological comfort in applying the TOR protocol. Further research should focus on this area to ascertain the effect after a run-in period. Other healthcare systems that wish to develop a similar training program should also tailor it to their local and cultural practice.

## Figures and Tables

**Figure 1 ijerph-18-01050-f001:**
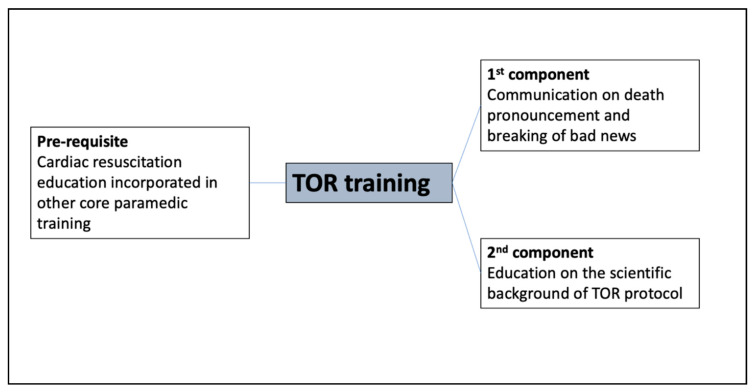
Pre-requisite and components of termination of resuscitation (TOR) training.

**Figure 2 ijerph-18-01050-f002:**
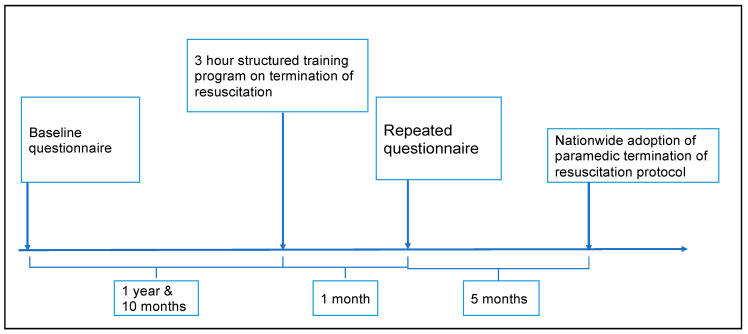
Chronology of events. Figure is not drawn to scale.

**Figure 3 ijerph-18-01050-f003:**
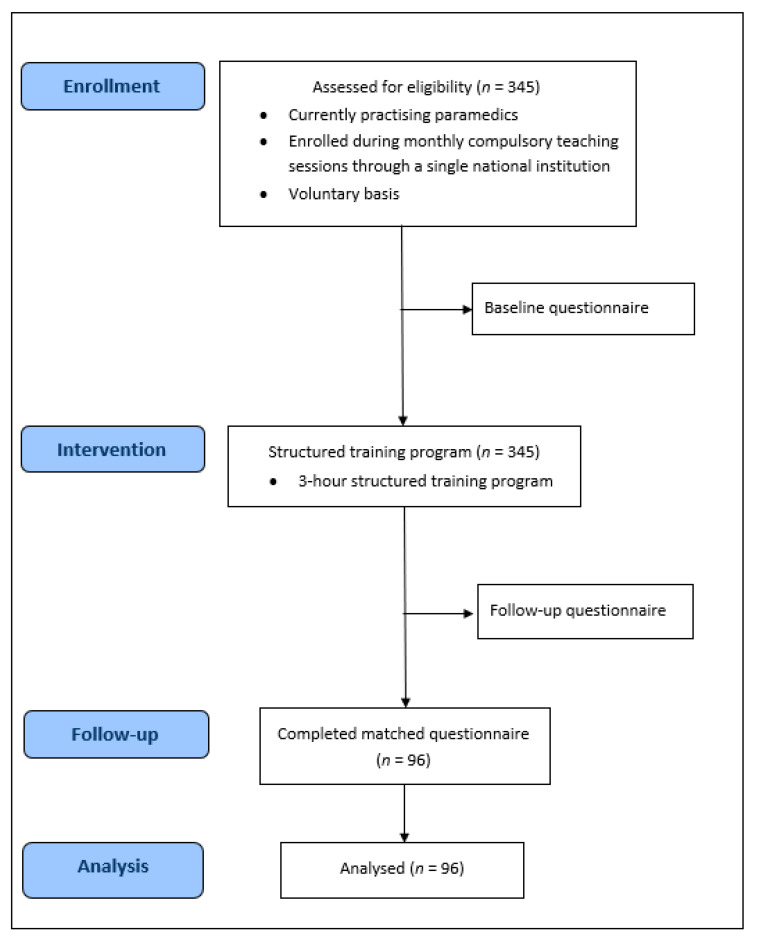
TOR study flow diagram.

**Figure 4 ijerph-18-01050-f004:**
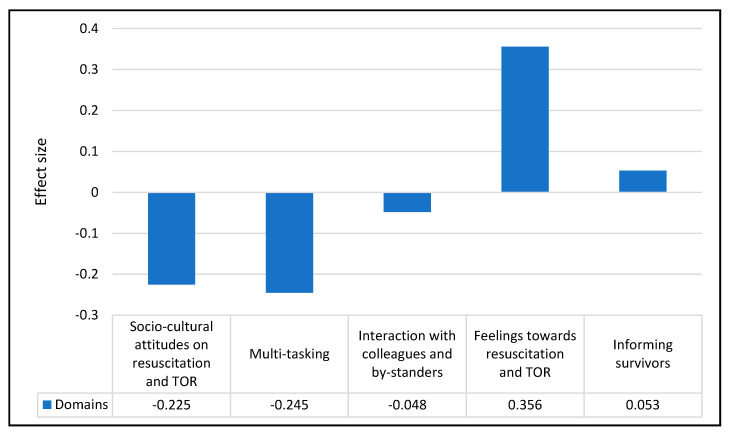
Effect size of the domains.

**Table 1 ijerph-18-01050-t001:** Training with targeted domains.

No	Item	Time (min)	Format	Remarks	Targeted Domains
1.	Methods on breaking bad news	15	Lecture—online	Didactic: completed prior to face-to-face training session	Sociocultural attitudes on resuscitation and TOR Informing survivors
2.	Film on the execution of TOR protocol: 1st scenario—Easy, routine execution 2nd scenario—Difficult execution with a stressed relative	20	Video—online	Didactic: completed prior to face-to-face training session	Sociocultural attitudes on resuscitation and TOR Informing survivors Interaction with colleagues and bystanders
3.	Lecture addressing rationale, the scientific basis of protocol, workflow, frequently asked questions and direct question and answer	60	Lecture	Face-to-face	Feelings towards resuscitation and termination of resuscitation
**4.**	Group learning practice on three different TOR scenarios 1st scenario—Exiting the protocol for obviously demised patients 2nd scenario—Easy, routine execution 3rd scenario—Difficult family where the execution of protocol is contra-indicated	85	Role-play simulation	Group learning comprising 4–6 paramedics and one facilitator	Sociocultural attitudes on resuscitation and TOR Informing survivors Multi-tasking

**Table 2 ijerph-18-01050-t002:** Respondents’ characteristics.

Participant Characteristics	Study Sample (*n* = 96)
Age in years, mean (SD)	31.1 (6.7)
Male, *n* (%)	49 (51.0)
Professes a religion, *n* (%)	82 (85.4)
Experience as a paramedic in years, mean (SD)	8.0 (5.9)
Government-employed, *n* (%)	71 (74.0)

Prior training received, *n* (%)	
Preceptorship	19 (19.8)
Documentaries, TV, videos, etc.	18 (18.8)
Personal reading	28 (29.2)
Religious training	12 (12.5)
Personal experience with death and dying, *n* (%)	76 (79.2)
Number of out-of-hospital cardiac arrest cases in the past year, *n* (%)	
0	5 (5.2)
1–5	32 (33.3)
6–10	26 (27.1)
≥11	33 (34.4)
Number of times death was pronounced (out-of-hospital) in the past year, *n* (%)	
0	3 (3.1)
1–5	58 (60.4)
6–10	18 (18.8)
≥11	17 (17.7)

**Table 3 ijerph-18-01050-t003:** Baseline and post-training psychological comfort total percentage in the various domains and the paired difference (*n* = 96).

Domains.	Level of Training Emphasis	Baseline Mean (SD) in %	Post-Training Mean (SD) in %	Paired Difference(95% CI)	*p* Value
1. Sociocultural attitudes on resuscitation and TOR	High	58.91 (18.54)	54.74 (19.48)	−4.17 (−8.59, 0.26)	0.065
2. Multi-tasking	Low	60.36 (16.78)	56.25 (17.95)	−4.11 (−7.82, −0.41)	***0.030**
3. Interaction with colleagues and bystanders.	Moderate	69.80 (15.37)	69.06 (18.46)	−0.74 (−4.98, 3.50)	0.731
4. Feelings towards resuscitation and termination of resuscitation	High	51.88 (13.43)	56.65 (10.85)	4.77 (1.42, 8.13)	**0.006**
5. Informing survivors	High	58.20 (15.74)	59.04 (15.10)	0.84 (−2.45, 4.12)	0.614
Overall domain mean	NA	59.82 (10.84)	59.10 (11.45)	−0.72 (−3.28, 1.83)	0.576
Overall PCT score as a percentage	NA	60.5 (8.36)	59.92 (8.73)	−0.58 (−3.24, 2.15)	0.668

PCT: psychological comfort total item percentage ranges from 0 to 100%, with a higher percentage indicating greater comfort. The paired difference is obtained by deducting the pre-training PCT % from the post-training PCT %. A negative value in paired difference indicates a drop in psychological comfort level post-training. * *p* value for multi-tasking domain in Wilcoxon signed-rank test >0.05.

## Data Availability

The data presented in this study are available on request from the corresponding author. The data are not publicly available due to privacy.

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
