# Peer review of "Improving Psychological Comfort of Paramedics for Field Termination of Resuscitation through Structured Training"

_ijerph, 2021, doi:10.3390/ijerph18031050_

Round 1

Reviewer 1 Report

The authors present results of a statistical analysis which was conducted after a training programme aiming to implement a new protocol on termination of resuscitation. The submitted study aims to address an important part of routine work in the emergency services. However, it also fells short in addressing the content of the training, details about statistical analyses and why the training does not result in a statistical significance when pre and post questionnaire answers compared. More specific comments are below.

Why did the authors perform t-test without assessing the distribution of the samples? There might be a need for a non-parametric test to evaluate pre and post questionnaire answers. Therefore, I would suggest authors do the Shapiro-Wilk test to evaluate the normality of the data and then perform a t-test or Mann-Whitney U test.

The training programme fells short to result in a statistical significance when pre and post questionnaire answers compared as the authors also mention in the Results and Discussion sections. What can be potential reasons for this? Why the authors did not also perform statistical analyses to compare differences for the factors such as gender, age, work experience or workload? This may provide some answers to this question.

The authors do not provide details about the contents of the training programme. For instance, how role-playing simulation was prepared and applied? Did the role-playing include only paramedics or also actors as well? Did it simulate different scenarios which can make the paramedics uncomfortable or only one type of scenario?

Minor comment:

Page 3, EMS was used before defining the abbreviation for Emergency Medical Service. First, it should be defined and then abbreviation can be used.

Reviewer 2 Report

This study analyses the impact of structured training on the psychological comfort paramedics before the implementation of a prehospital TOR protocol in Singapore. The work is in the scope of the journal, however, redaction and structure should be improved as indicated below, especially the methods should be clearer; the author is recommended to identify and practice sophisticated objectives for a journal publication. The author must justify the following points:

Comment 1: I couldn't identify the novelty of the paper. Hence, the paper should be revised to highlight novelties. Please consider that this lack of novelty starts with the Abstract, Introduction, and Conclusion. Furthermore, it is highly important to present a sufficient scientific contribution to this work.

Comment 2: The author is using (we / our) too much. Please consider that this is a scientific journal publication, where you need to avoid some phrases like (we, our, ….). Instead, you can use (this work, this study, this analysis….).

Comment 3: The word “Methodology” should be “methods” or “Materials and Methods”, the word methodology is the study/analysis of methods and should only be used when addressing epistemologies/ontologies https://en.wiktionary.org/wiki/methodology#Usage_notes. Besides, the methods are not outlined with the necessary vigor. This section needs to be reorganized and highly developed to clarify all steps of the proposed analysis. Hence, a suggestion to answer this point can be by adding a figure that shows the performance analysis and development of this work in a way to facilitate the understanding process for readers.

Comment 4: The author must unify the citation form for the whole manuscript including the reference section.

Comment 5: In line 122, the author mentioned the Morrison Questionnaire for the first time in this work, without highlighting this questionnaire in the Introduction. The author must know that each publication is individual and separated. Hence, this is not the right place to present this questionnaire.

Comment 6: Section Methodology, is not outlined with necessary vigor. The author needs to include sufficient methodological details in the paper and elaborate on the produced results from the proposed methods. The paper should provide enough information to be replicable by other researchers. Some sections must be relocated and rewritten to make it clearer for the readers. Did the author conduct an interview or questionnaire? Where is the list of questions used? It would be helpful for the reader to include the conducted interview or questionnaire as a supplementary file. The most important issue to be justified in this work is to illustrate and clarify how the proposed materials and methods presented in Section (methodology) have been conducted to collect the output results? Data collection and Data analysis subsections are very concise. These two parts of the study must show the development of the analysis of this work. This could help understanding how the results were obtained. Results and Discussion sections would be helpful to be discussed using more figures and tables to make the work citable by other researchers.

Comment 7: The Conclusion section is very concise and missing lots of necessary details. For example, the author needs to highlight the novelty and the materials and methods used in this work, and point out the collected results. Then, present a summary of the limitations of this research as well as the recommendation for future works. 

Round 2

Reviewer 1 Report

The authors responded to all the reviewer comments. The manuscript was improved significantly with additional data and information. The submitted study can be accepted as a publication.

Reviewer 2 Report

The work has greatly developed, where the author met all my comments.